# Surface Topography of Thermoplastic Appliance Materials Related to Sorption and Solubility in Artificial Saliva

**DOI:** 10.3390/biomimetics9070379

**Published:** 2024-06-23

**Authors:** Liliana Porojan, Flavia Roxana Toma, Mihaela Ionela Gherban, Roxana Diana Vasiliu, Anamaria Matichescu

**Affiliations:** 1Department of Dental Prostheses Technology (Dental Technology), Center for Advanced Technologies in Dental Prosthodontics, Faculty of Dental Medicine, “Victor Babeș” University of Medicine and Pharmacy Timișoara, Eftimie Murgu Sq. No. 2, 300041 Timisoara, Romania; flavia.toma@umft.ro (F.R.T.); roxana.vasiliu@umft.ro (R.D.V.); 2National Institute for Research and Development in Electrochemistry and Condensed Matter, 300569 Timisoara, Romania; mihaelabirdeanu@gmail.com; 3Department of Preventive, Community Dentistry and Oral Health, Center for Advanced Technologies in Dental Prosthodontics, Faculty of Dental Medicine, “Victor Babes” University of Medicine and Pharmacy Timișoara, Eftimie Murgu Sq. No. 2, 300041 Timisoara, Romania; matichescu.anamaria@umft.ro

**Keywords:** thermoplastic appliance, artificial saliva, sorption, solubility, surface topography

## Abstract

(1) Background: PETG (polyethylene terephthalate glycol) is a transparent, inexpensive, and versatile thermoplastic biomaterial, and it is increasingly being used for a variety of medical applications in dentistry, orthopedics, tissue engineering, and surgery. It is known to have remarkable properties such as tensile strength, high ductility, and resistance to chemical insults and heat, but it can be affected by various environmental conditions. The aim of the present study was to evaluate the topographical characteristics of four thermoplastic dental appliance materials in relation to water sorption in simulated oral environments (artificial saliva samples with different pH values). (2) Methods: The following four types of PETG clear thermoplastic materials were selected for the present study: Leone (L), Crystal (C), Erkodur (E), and Duran (D). In relation to the desiccation and water-uptake stages, their water sorption (Wsp) and solubility (Wsl) were calculated, and the surface topographies were analyzed on two length scales. The surface roughness was determined using a contact profilometer, and nanoroughness measurements were generated by three-dimensional profiles using an atomic force microscope (AFM). Statistical analyses (one-way ANOVA and unpaired and paired Student *t*-tests) were performed. (3) Results: After saliva immersion, the weights of all samples increased, and the highest sorption was recorded in a basic environment. Among the materials, the water uptake for the L samples was the highest, and for E, it was the lowest. In relation to water solubility, significant values were registered for both the L and C samples’ materials. After immersion and desiccation, a decreasing trend in microroughness was observed. The AFM high-resolution images reflected more irregular surfaces related to saliva immersion. (4) Conclusions: The sorption rates recorded in water-based artificial saliva were higher for basic pH levels, with significant differences between the samples. There were also significant differences related to the behaviors of the materials included in the study. In relation to roughness, on a microscale, the surfaces tended to be smoother after the saliva immersions, and on a nanoscale, they became more irregular.

## 1. Introduction

Thermoplastic materials intended for orthodontic appliances comprise polymers with different characteristics. Their clinical efficiency may be affected by different factors, and these may be related, on the one hand, to the materials’ properties and, on the other, to the oral environment [1,2,3]. The polymers most commonly used to manufacture orthodontic aligners are polyethylene terephthalate (PET), polyethylene terephthalate-glycol (PETG), polyethylene vinyl acetate (PEVA), polypropylene (PP), polycarbonate (PC), polyurethanes (TPUs), and polymer blends.

Polyethylene terephthalate (PET) is a thermoplastic polymer that was patented in 1941 and has proven its viability as a versatile and durable material in different non-medical and medical fields. Using further iterations of PETs, a new polymer has been created—PETG (polyethylene terephthalate glycol)—which is the result of the copolymerization of PET with CDHM (cyclohexanedimethanol) in place of ethylene-glycol in the material’s backbone [4]. PETG is a non-crystallizing amorphous copolymer of PET.

Depending on their molecular structures, thermoplastic polymers such as PET can be classified into amorphous or semicrystalline polymers. The amorphous form has an irregular crystal structure and is transparent, more aesthetic, and softer, and it has more proven ductility; therefore, it can be bent and stretched without breaking, with better impact resistance. The semicrystalline form has both amorphous (irregular) and crystalline (uniform) molecular structures. It is opaque and white and demonstrates greater mechanical strength, rigidity, and hardness, with good chemical resistance. The glycol group modifies the original PET from a semicrystalline form to a more amorphous form in PETG, improving its mechanical and aesthetic (transparency) properties [1,5,6,7].

Due to this metastable structure, the material has significant characteristics such as considerable tensile and strength properties, viscoelasticity and flexibility, ductility, shape memory, reduced crystallization capacity when applying heat, improved heat and chemical resistance, and high transparency [1,4,8]. Its melting point temperature (Tm) is between 245 and 265 °C, and the glass transition temperature (Tg) of PETG is 85 °C, which signifies the temperature at which the material becomes soft and supple, making it easier to manipulate [9,10]. These values allow better tolerance to increased temperatures compared to other thermoplastic polymers, and this material has been found to maintain its shape memory in the intraoral temperature range (up to 57 °C) [11,12,13,14].

However, it has been observed that PETG is influenced by high temperatures, water, the chemical nature of the environment, organic solvents, morphologies, thicknesses, manufacturing processes and finishing techniques, and the times elapsed after elastic deformations [15,16]. Further, it may have a tendency toward increased internal stress, wear, and potential breakage [17]. Thus, exposure to the oral environment could induce changes in the properties of this material and compromise the effectiveness of treatment [18].

In the oral environment, materials are subject to mechanical fatigue and hydrothermal degradation (ageing) [19]. The long-term use of a retainer induces discoloration and alteration of its optical properties, and the surface morphology shows distortions, microcracks, biofilm formation, and increased roughness [20,21]. Therefore, to maintain adequate strength during orthodontic treatment, an appliance must be replaced every 2 weeks [22].

It is known that all plastics can be affected by mechanical and environmental factors, leading to the release of small particles—secondary microplastics (MPs)—which arise from the physical, biological, and/or chemical fragmentation of plastic objects during their use. They end up in the environment as pollutants and have attracted attention in the scientific community due to their toxicity and distribution [23,24].

To date, various studies have been conducted to assess the stability of these materials in terms of water sorption; aging; and mechanical, colorimetric, and chemical changes through wear and exposure to saliva, as well as exposure to pigmented beverages during wear, which could affect the force transmission capacity and effectiveness of treatment [25,26].

PETG materials can be processed by different technologies, such as thermoforming, pressure molding, injection molding, and 3D printing. Vacuum-thermoformed appliances are widely used and suitable for removable tooth aligners and mouth guards, orthodontic retainers, and periodontal and temporomandibular joint splints [27,28]. Vacuum-formed appliances have grown in popularity due to advantages such as greater patient comfort, improved aesthetics, ease of manufacture, and lower cost [29]. In order to thermoform the supplied discs, it is necessary to use vacuum forming equipment that assures appropriate conditions of temperature, vacuum, and time in order to avoid failures such as air inclusions, porosity, and thermal degradation [5,30].

Thermoplastics are slightly branched or linear polymers with a high molecular weight, characterized by strong intramolecular covalent bonds associated with weak intermolecular Van der Waal’s forces that easily melt with increasing temperature, causing the polymers to overflow. Then, during cooling, the molecular chains solidify into new forms. This process of softening by heating and hardening after cooling can be repeated. The ability of thermoplastic polymers to adapt to a given model during the thermoforming procedure is a relevant characteristic of an aligner material [31].

Several studies have shown that the hardness, thickness, and transparency of the material may suffer a series of changes after undergoing the thermoforming process [14,32]. Ryu et al. [32] recorded changes in transparency and thickness for four different types of clear alignment materials, while the hardness of all four materials showed no significant difference compared to the values recorded before thermoforming. Bucci et al. [14] investigated two types of PETG aligners (one active and one passive), both with a thickness of 0.75 mm, and revealed that the thermoforming procedure changed the initial thickness of the material. However, they noted that the variation in thickness did not influence the clinical performance of the aligners.

An in vitro study [19] investigated eight different types of the most commonly used thermoplastic polymers in relation to water sorption and showed an increase in the overall thickness of all materials. Water sorption increases with time, with polyurethane alignment material (Invisalign) showing the highest values, followed by PETG.

Dalaie et al. [33] studied the resistance variation of thermoplastic materials to surface cracking (hardness, flexural modulus, thickness) of two PETG aligners with thicknesses of 0.8 mm and 1 mm, respectively, in contact with the oral environment for 14 days. Significant differences in hardness were observed only for the PET-G sheet 0.8 mm thick and were attributed to changes in the amorphous and crystalline structures or to the release of microplastics. Other experiments [34] showed that the contact of thermoplastic material with gingival epithelial tissues in a saline-based environment caused a disruption of cell membrane integrity, impaired cell metabolism, and cell-to-cell contact capacity. However, it was observed that this effect did not occur in artificial saliva.

When used for intraoral appliances, thermoplastic polymers should not release potential toxins that can cause adverse reactions. Therefore, there is a continuous need to test the cytotoxicity and to know the mechanical and physicochemical behavior of clear aligners produced by different manufacturers.

The purpose of our experiment was to supply objective evidence in order to support clinicians in making correct decisions when choosing a material for vacuum-formed appliances.

The aim of the present study was to evaluate the topographical characteristics of four thermoplastic materials used for dental appliances provided by different manufacturers in a simulated oral environment (artificial saliva with different pH values). The first null hypothesis is that sorption and solubility of PETG in artificial saliva are significantly affected by pH values. The second null hypothesis is that different commercial materials behave similarly in relation to sorption and solubility. The third null hypothesis is that the surface characteristics of PETG are affected by pH values. The fourth null hypothesis is that there are no significant differences between the four studied materials concerning surface properties in contact with saliva.

## 2. Materials and Methods

### 2.1. Specimen Preparation

The MINISTAR S pressure molding unit (Scheu-Dental, Iserlohn, Germany) was used for thermoforming the thermoplastic sheets. Discs with a thickness of 1 mm made from four types of PETG clear thermoplastic materials were selected for the present study: Leone (Leone SpA, Firenze, Italy) (L), Crystal (Bio Art Dental Equipment, Sao Carlos, Brazil) (C), Erkodur (Erkodent, Pfalzgrafenweiler, Germany) (E), and Duran (Scheu-Dental GmbH, Iserlohn, Germany) (D). Each sheet was heated by proximity to the heat source (220 °C) of the device—for 30 s, until they became malleable; then, they were pressed and vacuumed (under 4 bars of pressure) over the mold positioned in the pallet container. The cooling time was 60 s.

The gypsum mold was manufactured with 40 × 10 mm surfaces at an angle of 45° to the horizontal, and the thermoformed sheets were cut into square pieces with equal sides of 10 mm. The final thickness after thermoforming was different, depending on the material used. A total of 30 pieces of each material (total of 120) were prepared and evaluated. All specimens were placed in an airtight glass container with silica gel, stored in an incubator at 37 °C, and evaluated as control samples (Lc, Dc, Ec, and Cc).

The samples were then divided into three subgroups (Figure 1) and deposited in glass recipients containing artificial saliva, as follows: 10 pieces of each material were stored in saliva with neutral pH = 6.7 (first subgroup L0, D0, E0, and C0), in saliva with basic pH = 8.3 (second subgroup L+, D+, E+, and C+), and acidic pH = 4.3 (third subgroup L−, D−, E−, and C−), respectively, and evaluated after 14 days.

Modified Fusayama artificial saliva solution was prepared [g/L]: Urea 1.0, CaCl_2_·2H_2_O 0.795, NaH_2_PO_4_·H_2_O 0.690, NaCl 0.4, KCl 0.4, KSCN 0.3, Na_2_S·9H_2_O 0.005. All reagents used were employed as purchased with no further purification and dissolved in bidistilled water. The pH of the saliva was adjusted to the selected values (neutral = 6.7, basic = 8.3, and acidic = 4.3) by using NaOH and HCl [35].

In the last stage of the study, the specimens were re-dried in a desiccator at 37 °C with silica gel for 7 days.

### 2.2. Water Sorption and Solubility

Both water sorption (Wsp) and water solubility tests (Wsl) were carried out according to ISO 20795-2:2013 [36]. Samples from each material were dehydrated in a desiccator at 37 °C in glass containers with silica gel. After each stage, they were weighed only once using an analytical balance Kern ABT 100–5 NM (KERN & SOHN GmbH, Balingen, Germany), accurate to 0.00001 g.

Drying and evaluation cycles were performed until a constant mass (conditioned mass—m_1_) was obtained. Then, the samples were immersed in artificial saliva with different pH values and were hydrated until a constant mass was obtained (water saturation—m_2_). Next, they were stored in a desiccator to be dehydrated again until a constant mass (reconditioned mass—m_3_) was obtained.

The Wsp and the Wsl values were achieved by using Equations (1) and (2), respectively.
Wsp = (m_2_ − m_3_)/V  [µg/mm^3^](1)
Wsl = (m_1_ − m_3_)/V  [µg/mm^3^](2)
where m_1_ = constant mass of dehydrated samples; m_2_ = constant mass of hydrated samples; m_3_ = constant mass of reconditioned samples, the values being expressed in µg; and V = volume, in mm^3^ [32].

### 2.3. Surface Roughness Measurements

Surface roughness was assessed using a contact profilometer with a 2 µm diamond stylus (Surftest SJ 201-Mitutoyo, Kawasaki, Japan). The mean value of three measurements recorded in three randomly chosen areas of a surface was calculated. The values Ra (µm), arithmetic average roughness, and Rz (µm), maximum absolute vertical roughness, were obtained. The sampling length was 0.8 mm at an applied force of 0.7 mN. The evaluation was carried out after each stage: dehydration, hydration, and reconditioning through drying for all four tested materials and related to different pH values of the artificial saliva.

### 2.4. Atomic Force Microscopy (AFM)

Nanosurface topographic characterization of the samples was investigated with an atomic force microscope (Nanosurf EasyScan 2 Advanced Research-Nanosurf AG, Liestal, Switzerland). The values for Sa (nm), average nanoroughness, and for Sy (nm), maximum amplitude of heights, were recorded. AFM provided three-dimensional profiles of the surfaces (2.2 × 2.2 µm^2^) in non-contact mode. Measurements were made for the control, hydration, and re-drying stages for different materials and pH values of the artificial saliva.

### 2.5. Statistical Analysis

Statistical analysis was performed using IBM SPSS Statistics 21 software (IBM, New York, NY, USA). By applying the Shapiro–Wilk test for distribution, we found that the data were normally distributed in most cases (*p* > 0.05); for a complete investigation, parametrical and non-parametrical tests were performed. One-way ANOVA test was applied for statistical analysis of several dependent groups (how material is affected by the pH of the saliva after the hydration or re-drying stage) and Tukey post hoc test for multiple pairwise comparisons of means.

Unpaired Student *t*-Test was performed for two varied groups (to compare two materials subjected to the same treatment within a stage), and paired Student *t*-Test was used for two-times moments circumstances (for comparing a material from the control group with the same material after saliva immersion and desiccation stage, respectively). For the present study, the level of significance was established at 0.05.

## 3. Results

### 3.1. Water Sorption and Solubility

Table 1 summarizes the average results with respect to water sorption (Wsp) and solubility (Wsl) for the four investigated materials immersed in artificial saliva with different pH values (neutral 0 = 6.7, basic + = 8.3, acidic − = 4.3) and reconditioned.

After immersion, the weight of all samples increased by 0.33–0.52% in neutral saliva (0), 0.69–1.41% in basic saliva (+), and 0.46–1.27% in acidic saliva (−). The highest sorption was recorded in the basic environment for all polymers; among materials, the water uptake for L (+, −) was the highest, and for E (+, −), it was the lowest. The water sorption behavior after immersion in artificial saliva with different pH (0, +, −) is shown in Figure 2a, and the water solubility is shown in Figure 2b.

Significant differences (*p* < 0.01) were observed regarding pH values via one-way ANOVA test and Tukey post hoc test for pairwise comparison of the materials, except (*p* > 0.05) between L+ and L−, D+ and D−, and C0 and C−. The unpaired Student *t*-Test was used to compare the water sorption of the materials within the same pH value (0, +, −), and it showed significant differences (*p* < 0.05). Regarding water solubility, significant values (*p* < 0.01) were registered for the pairwise comparison of the materials, and the same happened for L and C, mainly in basic and acidic saliva, when comparing two materials in the same pH. Statistical tests (Table 2) showed significant differences, both related to the pH values and between materials within the same pH value.

### 3.2. Surface Roughness Measurements

The average Ra (µm) and Rz (µm) values with SD (standard deviations) for control and hydrated specimens are summarized in Table 3 and shown graphically in Figure 3.

After immersion, a decreasing trend in roughness was observed, except for E+ and E− (Ra increased), which is the material with the lowest value of water sorption (Wsp). For C0 and D0 groups, with the highest water sorption values, the lowest roughness values were recorded. Similar values were obtained for Lc and L0, Ec and E0, D+ and D−, and E+ and E−.

The one-way ANOVA test (α = 0.05) reported significant differences (*p* < 0.05) related to the pH values for materials L, E, and C when the statistical analysis was performed to compare the roughness of the surfaces immersed in artificial saliva with different pH (0, +, −). Following pairwise comparison (Tukey post hoc test), there were significant differences between L0 and L−, L+ and L−, E0 and E+, E0 and E−, and C (all).

The average Ra (µm) and Rz (µm) values with the SD (standard deviation) for control and re-dried specimens are presented in Table 4 and Figure 4.

After reconstruction via desiccation, the surface roughness decreased for all materials, more obviously for the basic group, followed by the neutral group. The lowest Ra values were recorded for C0 and D0, the samples with the highest Wsp (water sorption) values. Similar values were reported for E0, E+, and E−.

After the desiccation stage, significant differences (*p* < 0.05) were obtained for L and D, as well as between pairs of values - L0 and L+, L+ and L−, D0 and D+, and D0 and D−.

*p*-values resulting from the statistical analysis of the one-way ANOVA test for hydration and re-drying stages are presented in Table 5:

The unpaired Student *t*-Test (Table 6) was applied to compare the roughness of the surfaces of the materials within the same pH value (0, +, −). The test revealed significant differences (*p* < 0.05) for L0 and D0, C0; for D0 and E0, E0 and C0; for L− and D−, E−, C−; for D− and E− after hydration; for L+ and D+, C+; and for D− and E− after reconstruction.

The paired Student *t*-Test (Table 7) was performed to compare control and one-stage samples for the same material, with significant differences being reported among Lc and L−, Dc and D0, D+, D−, Cc and C0, and C+ for the hydration stage; and Lc and L0, L+, Dc and D0, D−, Ec and E0, Cc and C0, C+, and C− for the re-drying stage.

### 3.3. Atomic Force Microscopy

The average Sa (µm) and Sy (µm) values with SD (standard deviations) for the control and hydrated specimens are summarized in Table 8 and shown graphically in Figure 5.

AFM three-dimensional high-resolution images show the scattered data throughout the measured sample with an area of 2.2 × 2.2 μm^2^ (Figure 6).

The control group showed a more smooth and uniform surface (with Sa values of 3.16–9.13 nm). The surfaces became less flat after water sorption; the Sa values increased by 4.48–6.69 times.

The average Sa (µm) and Sy (µm) values with SD (standard deviations) for control and re-dried specimens are summarized in Table 9 and shown graphically in Figure 7:

The surfaces are rougher after water sorption for a neutral pH, where the Sa values increase by 1.58 times, and the opposite happens for basic and acidic pH, where the Sa values decrease by 1.31–1.49 times. These aspects can be found in the three-dimensional representation (Figure 8).

The one-way ANOVA test reported significant differences (*p* < 0.05) regarding the pH values for all materials when the statistical analysis was performed to compare the nanoroughness of the surfaces after saliva immersion and reconditioned after immersion in artificial saliva with different pH values (0, +, −). Following pairwise comparison (Tukey post hoc test), significant differences between all pairs were confirmed. The unpaired Student’s *t*-Test was applied to compare the roughness of the surfaces of the materials within the same pH value (0, +, −). The test revealed significant differences (*p* < 0.05) between all pairs. The paired Student *t*-Test was performed to compare control and one-stage samples for the same material, with significant differences being reported among all pairs.

The mean grain size was calculated for each step, material, and immersion medium. A decrease in size can be observed both after hydration and desiccation (Figure 9), reflected in the decrease in the surface roughness. There is a medium positive correlation (coefficient of 0.36) between the grain size and surface microroughness.

## 4. Discussion

The daily wearing of thermoplastic appliances by patients leads to continuous contact between their surfaces and the oral environment. This fact generates concern about associated surface topographical alterations related to the exposure. The water sorption process in the polymers is considered multifactorial. It may occur in particular because of the hydrophilic nature of its polymer units, its amorphous and/or crystalline structure, and its intermolecular spaces and porosities. The hygroscopic expansion of polymeric materials may affect the adaptation of orthodontic aligners, resulting in changes in the orthodontic forces [37].

Water sorption and solubility can be correlated to a particular polymer molecular weight, density, chemical composition, and crystallinity of the tested materials, which is characterized by different water molecule penetration into the polymeric material. Water sorption and water solubility related to ISO specifications should not exceed 32 µg/mm^3^ and 5 µg/mm^3^, respectively [36,38,39].

In this study, significant (*p* < 0.05) sorption rates were recorded for all pH values, higher in the basic medium (mean values of 4.10 µg/mm^3^ for a neutral pH, 7.78 µg/mm^3^ for an acidic pH, and 9.48 µg/mm^3^ for a basic pH); among materials, the highest rates were found for L, and the lowest were found for E (mean values of 10.09 µg/mm^3^ for L, 7.20 µg/mm^3^ for D, 5.06 µg/mm^3^ for E, and 6.13 for C µg/mm^3^). After immersion, the weight of all samples increased by 0.33–0.52% in neutral saliva (0), 0.69–1.41% in basic saliva (+), and 0.46–1.27% in acidic saliva (−). Some studies reported a PETG water sorption of around 2.15–2.19% [40]; others reported rates of 0.8% [32] after 2 weeks, with a significant increase in the specific weight. Given that the Wsp and Wsl of PETG in artificial saliva were significantly affected by the pH of the medium and that the materials do not behave similarly in terms of sorption and solubility, the first null hypothesis is accepted, while the second one is rejected.

The solubility values were higher for L (a mean value of 7.66 µg/mm^3^) and C (a mean value of 8.08 µg/mm^3^); in general, materials with higher water sorption values did not demonstrate higher solubility. Negative values of water solubility may indicate that thermoplastic materials absorb water, which leads to an alteration in the microstructure and an increased weight, followed by an incomplete dehydration of the materials, as found in this study for D and E, in basic and acidic environments. They do not indicate that no solubility occurred in these materials but may suggest a low solubility [41].

The surface morphology of the materials is irregular, even when not exposed to the simulated oral environment. These characteristics may be attributed to the lack of control in the manufacturing process. Some studies have shown [20,21] increased roughness and discoloration after hydration. However, during this study, these irregularities tended to be less accentuated after saliva immersions on the microscale. The surface roughness decreased for all materials and pH mediums, except E+ and E− (close values), which showed an increase in values, being the samples with the lowest water sorption rates. For the other materials, it seems that the increase in water uptake influenced, to some extent, the decrease in roughness. After re-drying, the Ra values further decreased, more prominently for the basic pH, which is the environment where sorption was the highest. The surface showed low surface roughness, proving that the basic environment is the most favorable for PETG. Significant differences in mean roughness values were reported among Lc and L−, Dc and D0, D+, D−, Cc and C0, and C+ for the hydration stage as well as for Lc and L0, L+, Dc and D0, D−, Ec and E0, Cc and C0, C+, and C− for the re-drying stage. The third null hypothesis that the surface characteristics of PETG are affected by pH values was accepted. In the present study, the mean roughness values were between 0.075 and 0.117 µm for the hydration stage and between 0.07 and 0.108 µm for the re-drying stage; other studies reported mean values of the arithmetic average roughness Ra after water immersion between 0.09 and 0.14 μm [40].

The mean nanoroughness values increased after saliva immersion, with significant differences being found in the relationship with pH values for all materials, with all pairs related to the control group. The test which compared the roughness of the material surfaces within a pH value (0, +, −) revealed significant differences (*p* < 0.05) for L0 and D0, C0; for C0 and D0, E0; for L− and D−, E−, C−; for D− and E− after hydration; for L+ and D+, C+; and for D− and E− after reconstruction, the fourth null hypothesis thus being rejected.

Smoother and more uniform surfaces of the control group (with Sa values of 3.16–9.13 nm) were associated with a bigger grain size (61–156 nm). On the nanoscale, the surfaces became less flat after water sorption, the Sa values increased by 4.48–6.69 times, and the highest nanoroughness values were recorded for E+ and E− (close values). After re-drying, the surfaces became more uniform; the Sa values decreased for all materials and pH mediums. Compared to the control group, for a neutral pH, the Sa values remained higher and increased by 1.58 times, while the opposite happened for basic and acidic pH—the Sa values decreased by 1.31–1.49 times, possibly as a result of desiccation and alteration of the surface structure under the influence of the acidic or basic environment.

The surface topography of thermoplastic appliances at different length scales is of great interest because it directly influences key features like functional properties, durability, and optical parameters. Therefore, the clinical behavior is governed by the degree of unevenness of solid surfaces [42]. Sufficient knowledge regarding the degradation patterns of the materials and the identification of appropriate in vitro testing protocols enable correct strategies in order to make these materials more efficient [43], an important task being the development of in vitro simulations correlated to these patterns.

A limitation of this study can be represented by the difficulty of in vitro simulation for some of the parameters involved in the oral environment. Further studies will include the interaction of other parameters, like temperature changes and microbial accumulation, on the surface of the thermoplastic materials.

## 5. Conclusions

This in vitro study highlighted topographical alterations in thermoplastic PETG materials related to different pH values in a simulated oral environment. Within the limitations of this study, the following can be concluded:The water-uptake and sorption rates recorded in water-based artificial saliva were in normal standards for all pH values, with significant differences (the highest values were found for basic, followed by acidic values, and the lowest value was found for a neutral pH).The increase in water uptake influences, to some extent, the decrease in roughness.The basic environment (followed by the neutral one) is the most favorable for PETG materials—water sorption is increased, and surface roughness is reduced.On the microscale, the surfaces tend to be smoother after saliva immersion, while on the nanoscale, they become less flat. After desiccation, the microroughness remains low, and the nanoroughness decreases for acidic and basic pH and increases for a neutral pH.It is important to describe the surface topography on all length scales because it may influence the functional properties of the materials.

## Figures and Tables

**Figure 1 biomimetics-09-00379-f001:**
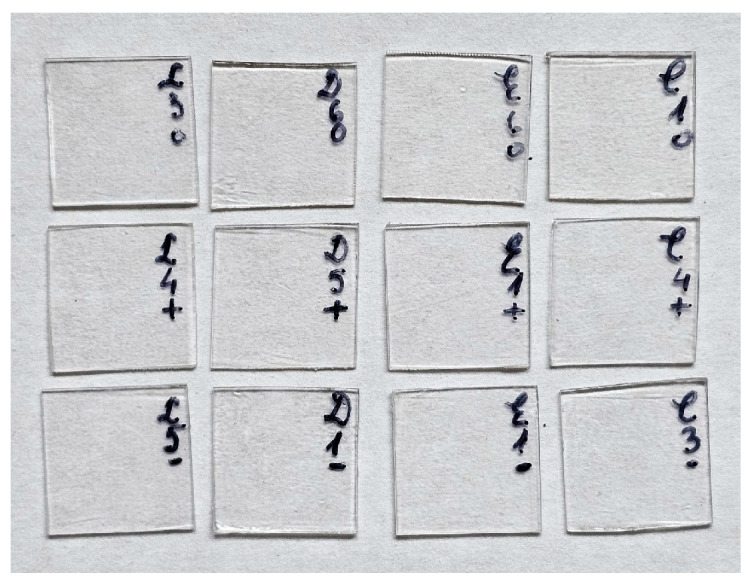
A selection of some samples of the investigated materials for three pH values; L = Leone, D = Duran, E = Erkodur, C = Crystal, pH: 0 = neutral, + = basic, − = acid.

**Figure 2 biomimetics-09-00379-f002:**
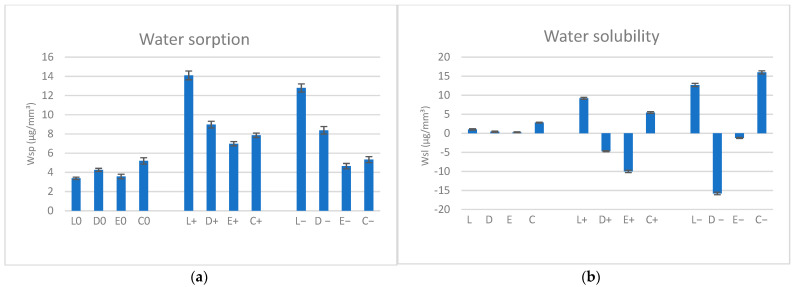
Water sorption and solubility (μm/mm^3^) of the investigated materials for three pH values: (**a**) water sorption, (**b**) water solubility; L = Leone, D = Duran, E = Erkodur, C = Crystal, pH: 0 = neutral, + = basic, − = acidic.

**Figure 3 biomimetics-09-00379-f003:**
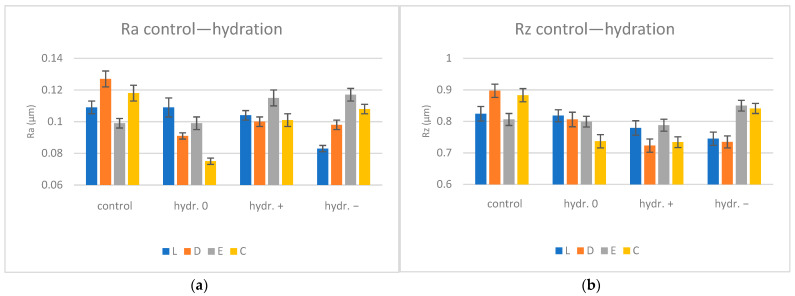
Profilometer surface roughness evolution for hydration conditions Ra (**a**) and Rz (**b**); L = Leone, D = Duran, E = Erkodur, C = Crystal, 0 = neutral, + = basic, − = acidic.

**Figure 4 biomimetics-09-00379-f004:**
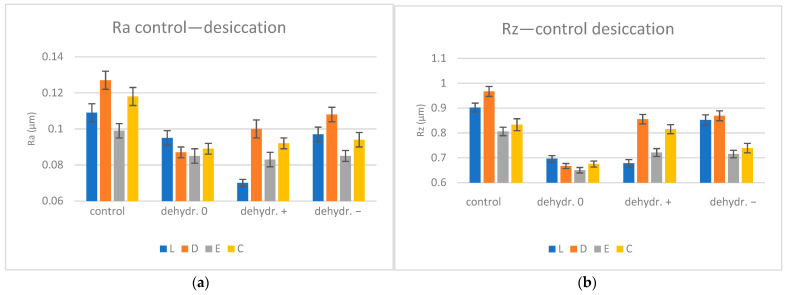
Profilometer surface roughness evolution for desiccation conditions Ra (**a**) and Rz (**b**); L = Leone, D = Duran, E = Erkodur, C = Crystal, 0 = neutral, + = basic, − = acid.

**Figure 5 biomimetics-09-00379-f005:**
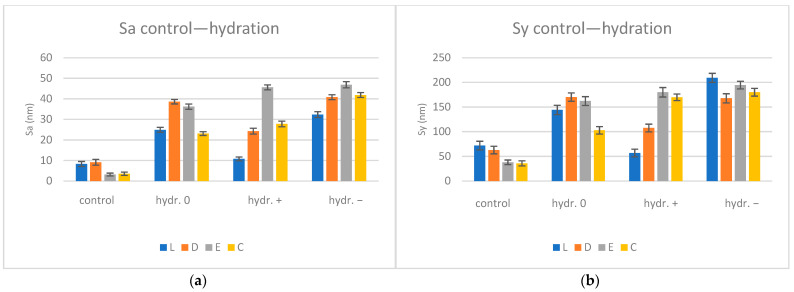
AFM surface roughness evolution for hydration conditions Sa (**a**) and Sy (**b**); L = Leone, D = Duran, E = Erkodur, C = Crystal, pH: 0 = neutral, + = basic, − = acidic.

**Figure 6 biomimetics-09-00379-f006:**
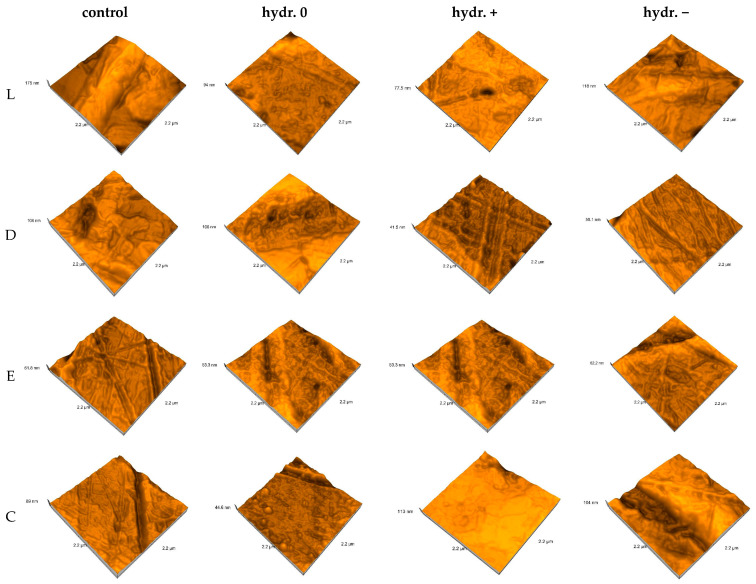
AFM surface topography evolution for hydration condition (L = Leone, D = Duran, E = Erkodur, C = Crystal), control group, and water-immersion group; 0 = neutral, + = basic, − = acidic pH.

**Figure 7 biomimetics-09-00379-f007:**
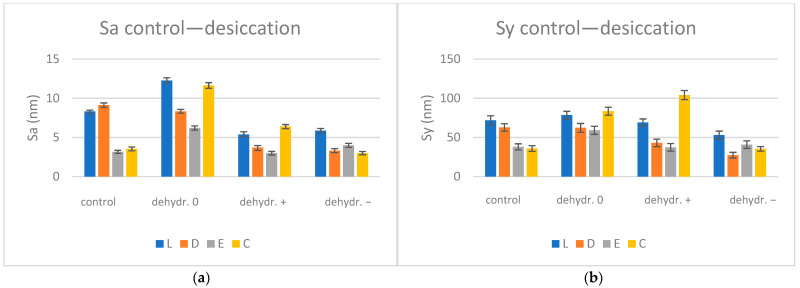
AFM surface nanoroughness evolution for desiccation condition Sa (**a**) and Sy (**b**); L = Leone, D = Duran, E = Erkodur, C = Crystal; 0 = neutral, + = basic, − = acidic.

**Figure 8 biomimetics-09-00379-f008:**
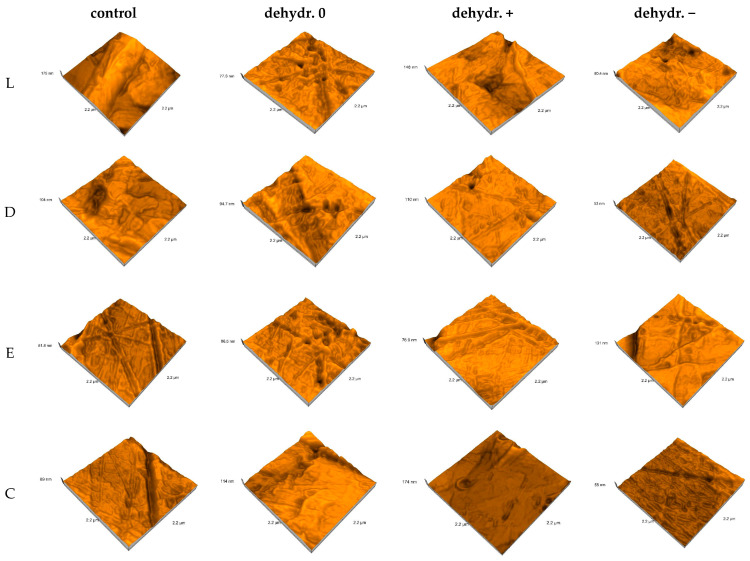
AFM surface topography evolution for desiccation condition (L = Leone, D = Duran, E = Erkodur, C = Crystal), control group, and re-dried group; 0 = neutral, + = basic, − = acidic pH.

**Figure 9 biomimetics-09-00379-f009:**
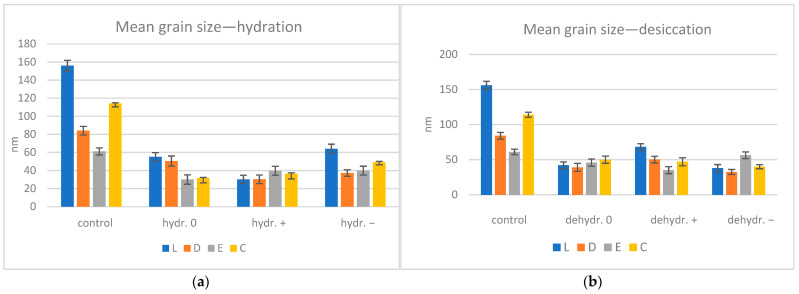
Graphical representation of the mean grain size (nm) for (**a**) control and hydrated groups, (**b**) control and re-dried groups; L = Leone, D = Duran, E = Erkodur, C = Crystal, 0 = neutral, + = basic, − = acidic.

**Table 1 biomimetics-09-00379-t001:** Wsp and Wsl values for the studied materials, in relationship with the pH value of the artificial saliva.

	pH_0_ = 6.7				pH+ = 8.3				pH− = 4.3			
	L0	D0	E0	C0	L+	D+	E+	C+	L−	D−	E−	C−
Wsp	3.38 ± 0.1	4.26 ± 0.1	3.57 ± 0.2	5.2 ± 0.3	14.1 ± 0.4	8.97 ± 0.1	6.98 ± 0.2	7.86 ± 0.2	12.7 ± 0.4	8.38 ± 0.3	4.65 ± 0.2	5.33 ± 0.3
Wsl	1.02 ± 0.1	0.44 ± 0.1	0.27 ± 0.3	2.8 ± 0.1	9.26 ± 0.2	−4.7 ± 0.1	−10 ± 0.2	5.46 ± 0.2	12.7 ± 0.1	−15.8 ± 0.1	−1.23 ± 0.1	16 ± 0.32

L = Leone, D = Duran, E = Erkodur, C = Crystal, 0 = neutral, + = basic, − = acid.

**Table 2 biomimetics-09-00379-t002:** *p* values (unpaired Student *t*-Test) for water sorption and water solubility.

*p*	L D	L E	L C	D E	D C	E C
Wsp 0	0.056	0.142	<0.001	0.102	0.031	0.001
Wsp +	<0.001	<0.001	<0.001	0.023	0.042	0.039
Wsp −	<0.001	<0.001	<0.001	0.011	0.027	0.14
Wsl 0	0.131	0.091	0.034	0.32	0.044	0.028
Wsl +	0.002	<0.001	0.028	0.049	<0.001	<0.001
Wsl −	<0.001	<0.001	0.045	0.011	<0.001	<0.001

L = Leone, D = Duran, E = Erkodur, C = Crystal, 0 = neutral, + = basic, − = acid.

**Table 3 biomimetics-09-00379-t003:** Average values of microroughness with SD for the control and saliva-immersed groups.

Material	Control		Hydr. 0		Hydr. +		Hydr. −	
	Ra	Rz	Ra	Rz	Ra	Rz	Ra	Rz
L	0.109 ± 0.004	0.824 ± 0.023	0.109 ± 0.006	0.818 ± 0.019	0.104 ± 0.003	0.779 ± 0.023	0.083 ± 0.002	0.745 ± 0.021
D	0.127 ± 0.005	0.897 ± 0.021	0.091 ± 0.002	0.806 ± 0.023	0.1 ± 0.003	0.723 ± 0.021	0.098 ± 0.003	0.735 ± 0.019
E	0.099 ± 0.003	0.806 ± 0.019	0.099 ± 0.004	0.799 ± 0.019	0.115 ± 0.005	0.788 ± 0.019	0.117 ± 0.004	0.85 ± 0.017
C	0.118 ± 0.005	0.883 ± 0.021	0.075 ± 0.002	0.737 ± 0.021	0.101 ± 0.004	0.734 ± 0.017	0.108 ± 0.003	0.841 ± 0.058

L = Leone, D = Duran, E = Erkodur, C = Crystal, Ra = Ra (µm)—arithmetic average roughness and Rz (µm)—maximum absolute vertical roughness.

**Table 4 biomimetics-09-00379-t004:** Average values of microroughness with SD for the control and re-dried saliva group.

Material	Control		Dehydr. 0		Dehydr. +		Dehydr. −	
	Ra	Rz	Ra	Rz	Ra	Rz	Ra	Rz
L	0.109 ± 0.005	0.902 ± 0.024	0.095 ± 0.004	0.696 ± 0.032	0.07 ± 0.003	0.678 ± 0.015	0.097 ± 0.004	0.852 ± 0.021
D	0.127 ± 0.005	0.967 ± 0.026	0.087 ± 0.007	0.667 ± 0.017	0.1 ± 0.005	0.855 ± 0.025	0.108 ± 0.004	0.869 ± 0.023
E	0.099 ± 0.004	0.806 ± 0.017	0.085 ± 0.004	0.65 ± 0.019	0.083 ± 0.004	0.721 ± 0.016	0.085 ± 0.003	0.715 ± 0.022
C	0.118 ± 0.005	0.833 ± 0.024	0.089 ± 0.003	0.675 ± 0.024	0.092 ± 0.003	0.815 ± 0.018	0.094 ± 0.004	0.739 ± 0.019

L = Leone, D = Duran, E = Erkodur, C = Crystal, Ra = Ra (µm)—arithmetic average roughness, and Rz (µm)—maximum absolute vertical roughness.

**Table 5 biomimetics-09-00379-t005:** *p* values (one-way ANOVA test) for the hydration and desiccation stages.

*p*	L0 L+ L−	D0 D+ D−	E0 E+ E−	C0 C+ C−
hydr.	0.001	0.382	0.041	<0.001
dehydr.	<0.001	0.048	0.966	0.826

L = Leone, D = Duran, E = Erkodur, C = Crystal, 0 = neutral, + = basic, − = acid.

**Table 6 biomimetics-09-00379-t006:** *p* values (unpaired Student *t*-Test) for the hydration and desiccation stages.

*p*	L D	L E	L C	D E	D C	E C
control	0.017	0.149	0.163	0.001	0.232	0.014
hydr. 0	0.031	0.243	<0.001	0.008	0.009	0.001
hydr. +	0.537	0.136	0.627	0.074	0.888	0.085
hydr. −	0.024	<0.001	<0.001	0.013	0.147	0.17
dehydr. 0	0.158	0.155	0.398	0.8	0.804	0.658
dehydr. +	0.002	0.146	0.017	0.04	0.368	0.32
dehydr. −	0.186	0.154	0.66	0.019	0.087	0.264

L = Leone, D = Duran, E = Erkodur, C = Crystal, 0 = neutral, + = basic, − = acid.

**Table 7 biomimetics-09-00379-t007:** *p* values (unpaired Student *t*-Test) for the hydration and desiccation stages.

*p*	Lc L0	Lc L+	Lc L−	Dc D0	Dc D+	Dc D−	Ec E0	Ec E+	Ec E−	Cc C0	Cc C+	Cc C−
c.-hydr.	1	0.272	<0.001	<0.001	<0.001	0.005	1	0.112	0.06	<0.001	0.019	0.186
c.-dehydr.	0.023	<0.001	0.102	<0.001	0.552	0.033	0.04	0.099	0.089	0.016	0.03	0.004

L = Leone, D = Duran, E = Erkodur, C = Crystal, 0 = neutral, + = basic, − = acid.

**Table 8 biomimetics-09-00379-t008:** Average values of nanoroughness (Sa, Sy) with SD for control and hydrated groups.

Material	Control		Hydr. 0		Hydr. +		Hydr. −	
	Sa	Sy	Sa	Sy	Sa	Sy	Sa	Sy
L	8.316 ± 1.2	71.889 ± 8.8	24.877 ± 1.2	143.99 ± 9.5	10.764 ± 0.9	56.778 ± 7.8	32.295 ± 1.4	209.15 ± 9.1
D	9.139 ± 1.4	62.814 ± 7.6	38.589 ± 1.1	170.11 ± 8.6	24.249 ± 1.4	107.39 ± 7.8	40.757 ± 1.2	167.61 ± 9.3
E	3.162 ± 0.7	38.028 ± 4.6	36.163 ± 1.3	162.08 ± 8.9	45.593 ± 1.2	179.87 ± 9.6	46.832 ± 1.5	194.42 ± 7.8
C	3.544 ± 0.8	35.855 ± 5.1	23.051 ± 0.9	102.75 ± 7.6	27.744 ± 1.4	169.64 ± 6.6	41.819 ± 1.2	179.99 ± 7.9

L = Leone, D = Duran, E = Erkodur, C = Crystal, Sa (nm)—average nanoroughness, and Sy (nm)—maximum amplitude of hights.

**Table 9 biomimetics-09-00379-t009:** Average values of nanoroughness (Sa, Sy) with SD for control and re-dried groups.

Material	Control		Dehydr. 0		Dehydr. +		Dehydr. −	
	Sa	Sy	Sa	Sy	Sa	Sy	Sa	Sy
L	8.316 ± 0.17	71.889 ± 5.8	12.26 ± 0.36	78.495 ± 5.8	5.389 ± 0.34	69.092 ± 4.6	5.872 ± 0.28	53.024 ± 5.1
D	9.139 ± 0.31	62.814 ± 4.8	8.309 ± 0.22	62.278 ± 4.8	3.709 ± 0.36	43.056 ± 4.7	3.316 ± 0.26	27.375 ± 3.6
E	3.162 ± 0.19	38.028 ± 3.9	6.205 ± 0.27	59.154 ± 3.9	2.984 ± 0.22	37.432 ± 4.9	3.99 ± 0.25	40.914 ± 4.8
C	3.544 ± 0.24	35.855 ± 3.6	11.631 ± 0.37	83.435 ± 3.6	6.39 ± 0.26	104.14 ± 5.8	2.996 ± 0.2	35.439 ± 2.9

L = Leone, D = Duran, E = Erkodur, C = Crystal, Sa (nm)—average nanoroughness, and Sy (nm)—maximum amplitude of hights.

## Data Availability

The original contributions presented in the study are included in the article; further inquiries can be directed to the corresponding authors.

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
