# Peer review of "Surface Topography of Thermoplastic Appliance Materials Related to Sorption and Solubility in Artificial Saliva"

_biomimetics, 2024, doi:10.3390/biomimetics9070379_

Round 1

Reviewer 1 Report

Comments and Suggestions for Authors

This in vitro study highlighted topographical alterations in thermoplastic PETG materials related to different pH values in simulated oral environment. The results are very interesting however a link between the different results is needed. The discussion of the results should be improved mostly the on the water sorption and solubility behavior. There are several questions that we find in the manuscript, which are listed as follow:

1.    Abstract: The abstract represents only an experimental description followed by some results. I believe that this kind of description does not work as an attraction for the reader!

2.    Introduction: The introduction provided a large list of references and studied systems. However, the introduction still lacks clarity. I would expect a more technical discussion based on the literature results to identify the main issues.

3.    Water uptake and sorption rates recorded in water based artificial saliva were above the standards for all pH values, with significant differences. The author's analysis of the water sorption and solubility results is only a description of the phenomenon. We should explore this mechanism to increase the intension.

Comments on the Quality of English Language

Moderate editing of English language required.

Reviewer 2 Report

Comments and Suggestions for Authors

Dear authors,

 Your work is devoted to the variation of artificial saliva pH on the behavior of 4 commercial PETG thermoplastic dental materials, concerning sorption and solubility processes in addition to surface roughness and topography.

Your results are presented in 5 tables, 8 figures and scientifically supported by 45 references.

In general, the manuscript is well divided and contains no errors.

Nevertheless, the starting four null hypotheses lack scientific support and Discussion and Conclusions items should be improved.

Here are some comments and suggestions in order to improve the quality of your paper.

Abstract

Line 22 - use curved instead of straight parenthesis (L) (C), etc.

Line 25 - …”uptake for L samples was the highest.

Line 26 - … were registered for both L and C samples materials

Introduction

Line 39-47: There is text repetition regarding the formation of PETG from PET material.

Line 48: “Due to this metastable structure, …

Line 51-56: You refer for PETG the two transition temperatures: melting and glass. What is their importance for this application? In you sample preparation (line 104) you employed a temperature of 220ºC for all PETG samples. What was the criterion for this value, that is, 135ºC above Tg and -25ºC below Tm ?

Materials and Methods

Line 120: …with no further purification and dissolved in bidistilled water. The saliva pH value was adjusted to the selected values by using …. -  please, included here the reagent used for this propose.

Line 127: Both water sorption (Wsp) and water solubility (Wsl) tests were

Line 137: The Wsp and the Wsl values were achieved by using the following equations (1) and (2), respectively.

  Wsp = (m2-m3)/ V    [mg/mm3]                        (1)

  Wsl = (m1-m3)/ V    [mg/mm3]                         (2)

where, m1 …..

Results

Line 180:  Table 1 summarizes the average results concerning ……

Then, you should include in this task the optical photographs of the 4 square samples under study (control). By this way you prove the samples were transparent. Moreover, through this simple characterization you can show whether or not there was an optical change in each of the steps: hydration and desiccation.

In addition, you should also add some bulk characteristics of the 4 commercial PETG samples, in particular their chemical composition. This is imperative. All in order to achieve a constructive discussion of the results, which is, in my opinion, the most unfavorable point of your paper.

Table 1: improve this table by adding a top line with the pH values, in accordance with the symbols used. You must also include the respective standard deviation.

Figure 1: improve this figure by,

1) adding the YY axis legend and an appropriated scale.

2) adding the limits allowed by ISO 20795-2:2013 - ref 26, i.e. 32 and 5 mg/mm3 for Wsp and Wsl, respectively.

3) with the above procedure, it will be possible to show that the samples studied in this work far exceed the threshold limits. How do you explain this behavior for these commercial samples? Nothing is mentioned in Discussion task.

Figure 2: improve this figure by adding the YY axis legend and an appropriated scale.

Legend of Fig2 – Profilometer surface roughness evolution for hydration conditions … (remove with SD)

Figure 3: improve this figure by adding the YY axis legend and an appropriated scale.

Legend of Fig3 – Profilometer surface roughness evolution for desiccation conditions … (remove with SD)

Figure 4: improve this figure by adding the YY axis legend and an appropriated scale.

Legend of Fig4 – AFM surface roughness evolution for hydration conditions … (remove with SD)

Figure 5: improve this figure. It has very low resolution and the scale is unnoticeable.

Legend of Fig5 – AFM surface topography evolution for

Figure 6: improve this figure by adding the YY axis legend and an appropriated scale.

Legend of Fig6 – AFM surface nanoroughness evolution for

Figure 7: improve this figure. It has very low resolution and the scale is unnoticeable.

Legend of Fig7 – AFM surface topography evolution for

Figure 8:  What do you mean by particle size in polymeric materials?  How was it quantified? Why no standard deviation is presented?

Line 260: Confirm the AFM area analyzed. Is it 2.2x2.2 mm ?

Discussion

In my opinion, it is impossible to discuss results without knowing the characteristics of these 4 commercial materials: similarities and differences.

First null hypothesis: What is your explanation for these materials having higher values for water uptake and sorption rates than those specified by the standards? Should they continue to be marketed? Basic pH is more disadvantageous, why? This is a contradiction in relation to human oral cavity. Include in your results discussion.

Second null hypothesis: Why don't samples L and C show negative water solubility values, regardless of saliva pH? Include in discussion of results.

Third null hypothesis: Which pH is more disadvantageous, acidic or basic? Which are the worst and best samples? Explain the meaning of irregular surface. References 19 and 20 point to discoloration and increased roughness. Are your results opposite to those reported by others authors? Include in your discussion.

Fourth null hypothesis: Neutral pH is the more disadvantageous concerning nanoroughness? Why? Is this an expected result for PETG materials? Which are the worst and best samples? Include in your discussion

Figures and tables should be included in this section in accordance with item 3 – Results, guiding the reader during discussion item.

Line 360-388 - this part of the discussion should be removed because it is not in line with the results obtained.

Finally, please, review all the paper and the English language too.

Good luck!

Comments on the Quality of English Language

Minor editing of English language is required.

Reviewer 3 Report

Comments and Suggestions for Authors

-Abstract

The aim of the study is stated as to evaluate the topographical characteristics of four  thermoplastic materials, without mentioning the sorption and solubility.

Results

I would suggest to cite the p values for statistically significant differences, or give a table showing the p values of the difference between study groups.

It would also better to have horizontal bars connecting the groups with statistically significant differences

-Discussion

Line 331:  "There is a low positive correlation.." Was there a correlation analysis done?

-Conclusions

"2. There is low positive correlation between water sorption and solubility"

I suggest to delete this statement as I did not see any correlation analysis (Like Pearson or Spearman correlation tests).

Comments on the Quality of English Language

-There are some orthography issues,e.g.,

Line 23: should be "...analyzed..."

Line 27: should be "...micro roughness..."

Line 30: should be "...differences...", etc

Line 31: I would suggest to use "included" instead of "taken into"

Line 32: should be "...smoother..."

Line 44: should be "...different..."

Line 49: should be "...considerable..."

Line 48-51. the font is not consistent

Line 57: should be "...processed..."

Line 70: should be "...crystalline..."

Line 91, 93: should be "...sorption..."

Line 91: is it artificial "saliva"?

Line 149: ...arithmetic...

Line 378: ..extremely...

Line 387: ...appropriate in vitro...

Line 390: ...develop...

Line 397: delete '...for...'

I would suggest to check the spelling and grammar

Round 2

Reviewer 2 Report

Comments and Suggestions for Authors

Dear authors,

- Your paper was greatly improved, but some changes were done without care. Thus, it is appropriate to review the manuscript again.

- I noticed that an error in the units of water sorption (Wsp) and water solubility (Wsl) completely reversed the main conclusion of your work, that is, from above the standards to in accordance with standards, regardless of saliva pH.

 - I also noticed that you added new references, including your recent work - Ref.41- for the same 4 PETG materials. Comparing these revised results (Wsp > 3 mg/mm3) with those previously published (Wsp < 3 mg/mm3 = 3000 mg/mm3) something is not right: you have a water uptake variation of 3 orders of magnitude!

- Assuming that those already published are correct, then those now obtained must be reviewed again. Check mass and volume units.

- And so, I ask again: What is your explanation for these materials having higher values for water uptake and solubility rates than those specified by the standards? Should they continue to be marketed?

- I reinforce once again: I don't know the meaning of particle size in amorphous bulk thermoplastics processed by thermoforming without additives. I do not know which particles sizes were evaluated by AFM and I recommend that these results be removed from the revised manuscript, unless you provide some explanation.

- Line 34: twice L sample

- Line 38: uptake and sorption ??

- Line 72: metastable

- Line 163 and 175: twice the objectives of the work

- Line 237: mg ??

- Line 403: irregular?  more rough or less flat

- Line 459: particle size ??

- Line 462: grain size ??

- Line 488: significantly higher ??

- Line 521:  more obviously ??

- Line 542: In materials science, the concept of grain size is different from particle size one.

- Figures 6 and 8 cannot be published as they are now. The AFM surface topography is still imperceptible.

Good luck!

Comments on the Quality of English Language

Read the paper again, this time more carefully.
